# Solid-state polymer electrolytes for high-performance lithium metal batteries

Snehashis Choudhury [1,3], Sanjuna Stalin [1,3], Duylinh Vu[1,3], Alexander Warren [1], Yue Deng[2], Prayag Biswal [1] & Lynden A. Archer [1]*

Electrochemical cells based on alkali metal anodes are receiving intensive scientific interest as potentially transformative technology platforms for electrical energy storage. Chemical, morphological, mechanical and hydrodynamic instabilities at the metal anode produce uneven metal electrodeposition and poor anode reversibility, which, are among the many known challenges that limit progress. Here, we report that solid-state electrolytes based on crosslinked polymer networks can address all of these challenges in cells based on lithium metal anodes. By means of transport and electrochemical analyses, we show that manipulating thermodynamic interactions between polymer segments covalently anchored in the network and "free" segments belonging to an oligomeric electrolyte hosted in the network pores, one can facilely create hybrid electrolytes that simultaneously exhibit liquid-like barriers to ion transport and solid-like resistance to morphological and hydrodynamic instability.

---

[1] School of Chemical and Biomolecular Engineering, Cornell University, Ithaca, NY, USA. [2] Material Science and Engineering, Cornell University, Ithaca, NY, USA. [3] These authors contributed equally: Snehashis Choudhury, Sanjuna Stalin, Duylinh Vu. *email: laa25@cornell.edu

Safe, cost-effective, and long-lasting electrical energy storage devices are essential to sustain progress in electrified transportation, mobile device technology, and autonomous machines including drones and advanced robotics. One highly sought-after pathway to such devices is through evolving today's lithium-ion batteries to so-called 'metal batteries' that replace the graphitic anode with an alkali metal block, including lithium, sodium and potassium[1,2]. These batteries are attractive because they offer the potential of augmenting the anode capacity by factors ranging from 3 to 10 and enable use of higher-energy conversion cathodes, including sulfur and oxygen[3–7]. Scalable approaches for overcoming fundamental challenges associated with morphological[8,9], chemical[10–13], and hydrodynamic[14] instabilities at the alkali metal anode have emerged in recent years to be crucial for progress. No electrolyte or electrochemical cell design presently exists that addresses all of these challenges.

Conventional wisdom holds that solid-state electrolytes composed of mechanically strong and chemically inert materials may offer a unified strategy for mitigating all sources of instability in a metal battery. The successes and failings of such all solid-state metal batteries, particularly those based on Li are beginning to emerge from fundamental[9,15–17] and application-focused studies[11,12,18,19]. Among the most important challenges include: (i) Difficulty in finding materials that simultaneously offer sufficient mechanical rigidity to slow the growth kinetics of non-planar metal deposits and at the same time provide fast room temperature bulk and interfacial ion transport—particularly when the alkali metal electrode is used in tandem with high-capacity intercalating cathodes such as $LiCoO_2$ (LCO), $LiNiMnCoO_2$ (NMC), and $LiNiCoAlO_2$ (NCA). (ii) Inability of many of the best electrolyte candidates to reversibly deform and flex to accommodate volume change at the anode. And, (iii) the complex, insulating interphases formed by solid-state electrolytes in contact with the reactive alkali metal electrode. Dense polyether based networks with high crosslink densities have also been reported in

multiple recent studies[18,20,21] to be effective in overcoming some of these challenges at low current densities. More recently, Wei, et al.[22] reported that liquid electrolytes that incorporate high molar mass polymers to form molecular entanglements in the liquid and thereby impart viscoelasticity are effective in stabilizing deposition of metals at intermediate current densities, particularly at electrodes composed of softer alkali metals such as sodium.

Here we report on the synthesis, physical and electrochemical properties of thin, solid-state polymer electrolytes formed directly on the surface of Li metal anodes. Using a facile light-initiated chemical reaction strategy we show that it is possible to create highly elastic, solid-state polymer interphases on Li that are able to flex and stretch to accommodate volume change at the anode during Li plating and stripping. An important finding is that the interactions and composition of such electrolytes can be tuned in the presence of a compatible liquid phase to create solid-state, elastic membranes that whether deployed as interphases or as fully solid-state electrolytes, impart high levels of reversibility to electrochemical processes at the anode. Such materials are shown for example to eliminate the hydrodynamic instability termed electroconvection by indefinitely extending the diffusion limited ion migration regime. Cycling studies in electrochemical cells composed of metallic Li anodes and commercial-grade nickel cobalt manganese oxide (NCM) cathodes further reveals that the membranes enable high coulombic efficiency (CE) and stable long-term cell operations, at substantially higher voltages than previously reported for ether-based electrolytes. Direct visualization of Li electrodeposition conclusively shows that the ability of the elastic interfaces formed in contact with the Li anode to promote compact deposition is an important source of the electrochemical stability and versatility of the materials.

## Results

**Physical Structure and Thermodynamic Characterizations.** Figure 1a shows a schematic of the synthesis procedure used to

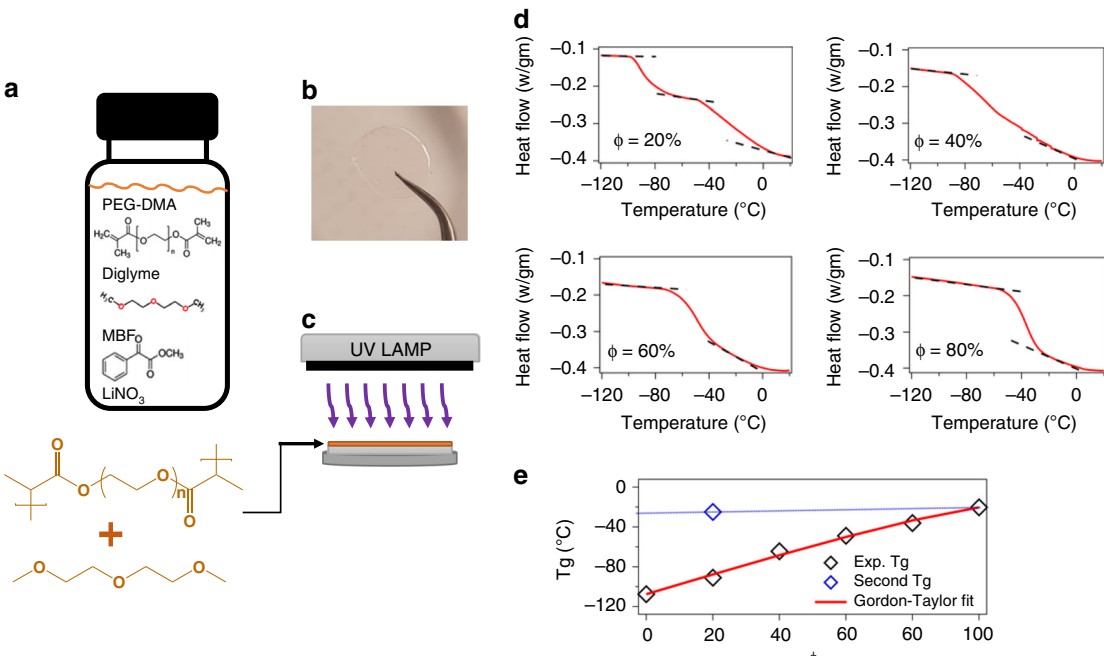

**Fig. 1** Physical Structure and Thermodynamic Characterization. **a** Chemical structures of the precursors used to synthesize polymer networks employed in the study. **b** Photograph of a typical polymer network specimen. **c** Schematic illustrating the concept of in situ crosslinking on a metal electrode. **d** Thermograms obtained from Differential Scanning Calorimetry (DSC) for polymer network membranes for various fractions (Φ) of PEGDMA. The dotted lines mark the step-change in the heat-flow discussed in the text. **e** Glass Transition temperature as a function of PEGDMA fraction (Φ) in the membranes. The red line is the Gordon-Taylor fit to the experimental results

create the solid-state polymer electrolytes used in the present study. It is a facile, single-step bulk polymerization process that does not require solvent. Specifically, poly(ethylene glycol) dimethacrylate (PEGDMA) of molecular weight 750 Da is added to Bis(2-methoxyethyl) ether (diglyme) at varying volume fractions. Both materials have good chemical stability with a Li metal electrode and, due to the well-known strong interactions between polyethers and methacrylates and between oligoethers and many lithium salts, spontaneously form low-viscosity, single-component liquid electrolyte mixtures in the absence of any solvent. We found that these mixtures can be easily coated on a Li substrate in a glovebox to form liquid films of almost any thickness desired on Li. Here, our interest is in proof of concept type demonstrations, so we focus only on a single thickness of approximately 100 μm. In the presence of a suitable free-radical initiator, such as methyl benzoylformate (MBF), it is also possible to induce each of the methacrylate groups to form up to two linkages with other PEGDMA chains, rapidly cross-linking and solidifying the coatings and trapping the mobile oligoether electrolyte component in the pores.

For simplicity a fixed ratio of lithium ions ($Li^+$) to ethylene oxide (EO) of 0.10 was maintained in the materials by addition of the lithium salt, Lithium Nitrate (LiNO3), which is widely reported to have beneficial effects in stabilizing the SEI formed at a Li metal anode. Crosslinking was achieved in the present work by exposing the mixtures to ultraviolet (UV) light ($\lambda = 320$ nm) to induce photopolymerization of the methacrylate groups. The result of the photopolymerization is a tough, elastic membrane approximately 100 μm thick, tightly bound to the underlying substrate. The degree of softness is dependent on the oligoether content in the overall mixture because only PEGDMA participates in the cross-linking reaction. Figure 1b shows a membrane with PEGDMA content ($\phi = 40\%$) that is transparent and homogenous, exhibiting no observable aggregates or signs of crystallite formation. Further it can be seen in Supplementary Fig. 1 that the material has a rubbery texture and is able to recover its structure fully even after large, macroscopic deformations. Because the cross-linking reaction is a bulk polymerization reaction, the membrane formation process can be carried out directly on a lithium metal electrode or on another substrate of lower surface energy (represented in the schematic of Fig. 1c). This versatility is important as it allows the materials to be studied in detail either in the form of intact solid electrolyte interphases on Li, as single-component solid-state electrolytes, or as freestanding films.

Evolution of the chemical bonding chemistry in the polymer electrolytes with varying PEGDMA content in the precursor solution was studied via Fourier transform infrared spectroscopy (FTIR). The results illustrated in Supplementary Fig. 2 show that the C=O bond[23] at ~1700 $cm^{-1}$ associated with PEGDMA increases in intensity and shifts to lower wave number with increasing PEGDMA content. This shift is an indication of a reduction in the effective moment of inertia of the absorbers, which correlates with a reduction in the spacing between network points in the cross-linked material. Thus, increasing PEGDMA content results in higher cross-link density, which leads to membranes that are macroscopically more elastic and mechanically stronger. Additionally, we find that the peak at ~1650 $cm^{-1}$, present in the uncrosslinked PEGDMA, significantly diminishes or disappears for all the crosslinked polymer samples indicating the PEG chains are highly crosslinked. Interestingly, we also find that at $\phi < 40\%$, the FTIR spectra in the "finger-print region" (<1000 $cm^{-1}$), shows multiple vibrational peaks in contrast to a single strong peak at higher $\phi$ values. We hypothesize that this transition is due to the confinement effect[24] on the free diglyme molecules due to the presence of a percolated PEGDMA network.

This can be rationalized from the fact that presence of multiple peaks in this region is an indication of the various modes of vibrations, while a single peak represents lack of freedom for vibrational relaxations.

We utilize Dynamic Mechanical Analyzer to understand the mechanical properties of the polymer network. In Supplementary Fig. 3, we observe that the tensile modulus linearly increases with the PEGDMA content. Importantly, this represents the increasing crosslinking density in the polymer samples as the content of crosslinked PEGDMA is higher. Supplementary Fig. 4 illustrates how such an in situ cross-linked membrane might be used as artificial solid-electrolyte interphases (ASEI) to inhibit physical and chemical instabilities at an alkali metal electrode. Before assessing the electrochemical consequences of this ASEI design, we first consider the fundamental physical features of the materials and on that basis elucidate their versatility. Results from differential scanning calorimetry (DSC) of the of pure diglyme—LiNO₃ ($\phi = 0\%$) as well as pure PEGDMA membrane-LiNO₃ ($\phi = 100\%$) are shown in Supplementary Fig. 5, while the DSC thermograms for intermediate $\phi$'s are provided in Fig. 1d. In the specific ranges of temperatures displayed in these figures, a second-order phase transition is observed that is associated with the glass transition of the polymer membranes. This transition reflects the phase change from melt to a glassy state due to kinetic entrapment of the molecules due to free volume reduction. The glass transition temperature (Tg) observed for $\phi = 0$, corresponding to pure diglyme-LiNO₃ and $\phi = 100\%$ (pure PEGDMA −LiNO₃) are −107.58 °C and −20.37 °C, respectively as shown in Supplementary Fig. 5.

It is known that compatible polymeric mixtures show distinct Tg's intermediate of the respective materials, while that of a fully-mixed blend result in a single Tg that is dependent on the molecular dispersion and relative strength of intermolecular forces in the materials. The membrane with 20% PEGDMA content shows two distinct Tg values of −91.01 °C and −25 °C (see Fig. 1d), which can be asserted to two-phases that are rich in diglyme and PEGDMA, respectively. Here, the membrane exists in two separate phases, which implies that the PEGDMA in the original mixture is not adequate to form a fully connected network after cross-linking. This means that only a small fraction of the added diglyme interacts with the cross-linked phase, while the rest exists as a second phase that is essentially a free liquid. Similar behavior in the Tg has been observed in partially miscible polymer blends that interact interfacially[25–27].

At $\phi = 40\%$, the glass transition event occurs over a visibly broadened range of temperature, indicating that the PEGDMA network in the membrane is at that point at the limits where it exists as a one single component. At this composition, the separate Tg values begin to merge, indicating a chemically homogenous material such that the PEG chains are permeated evenly throughout the membrane and it exhibits comparable mechanical toughness to a fully crosslinked network, while maintaining conductivity close to that of a neat (liquid) electrolyte. As the PEGDMA composition is increased further, the Tg values fully merge to a sharp transition (shown at $\phi = 60\%$ and above), indicating the formation of a single-phase percolated network. At this point, the cross-linked networks exhibit characteristics of so-called solid-state electrolytes: mechanically resilient yet limited by low ionic conductivity.

Figure 1e compares Tg values for the synthesized material at all PEGDMA content. The measured Tg values were fitted to the classical Gordon-Taylor relation:[28] $Tg = (w_1 Tg_1 + K_o w_2 Tg_2)/(w_1 + K_o w_2)$, where $w_1$ and $w_2$ are weight fractions of diglyme—LiNO₃ and PEGDMA-LiNO₃, while $Tg_1$ and $Tg_2$ are the glass transition temperature of the same. $K_o$ here is 0.35 that is obtained from the least square fitting of the experimental values,

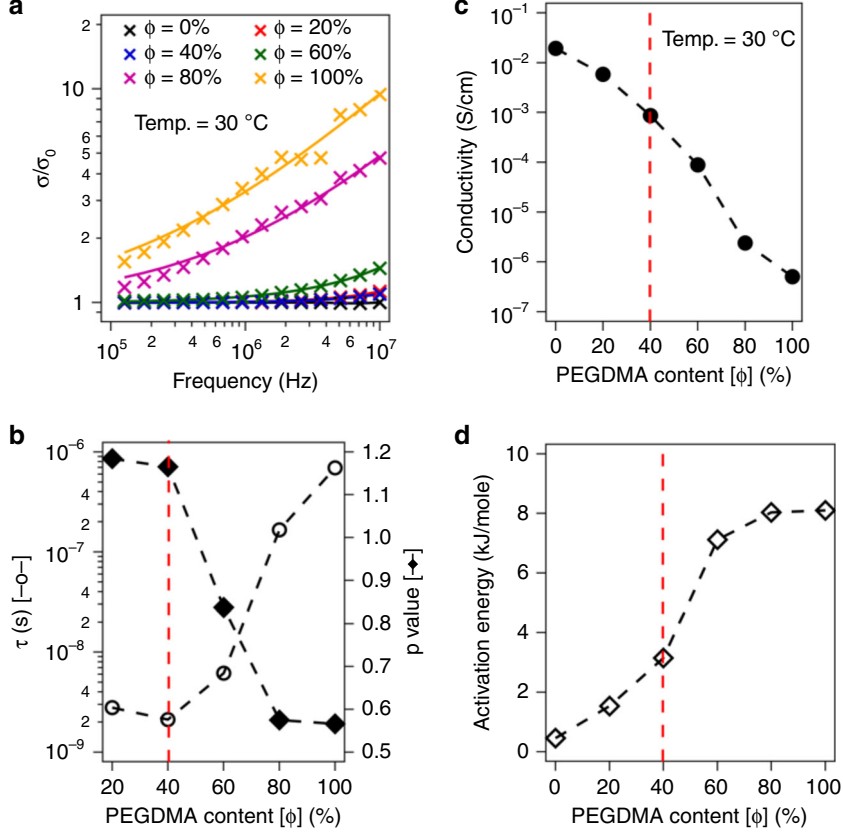

**Fig. 2** Structure-dependent Ion Transport Properties. **a** Normalized ac Conductivity as a function of frequency. Measured values are shown as markers, while lines represent fits using Cramer's Equation. **b** Results from frequency dependent conductivity measurements. Left axis shows the ion transport relaxation time and the right axis reports the ratio of the backward hop-rate to the forward hop-rate obtained from the power-law model. **c** Conductivity as a function of PEGDMA content at fixed temperature. **d** Apparent activation energy for ion transport obtained using the VFT model as a function of PEGDMA content

which is indicative of the relative change in heat capacities compared to the pure components. The Gordon-Taylor relationships holds for non-interacting ($\chi = 0$) polymer mixtures. It was noted that the second glass transition temperature at $\phi = 20\%$ that doesn't follow the Gordon-Taylor relation is close to the value at $\phi = 100\%$. Thus, it can be concluded that as the PEGDMA content increases, the first phase (rich in PEG network) continue to grow and essentially engulf the diglyme molecules to form a dense and swollen percolated network of polymer chains.

**Structure-dependent ion transport properties**. Frequency dependent conductivity can reveal important information regarding the ion transport mechanisms as well as structural arrangements. Figure 2a reports the normalized real component of a.c. conductivity as a function of frequency for different PEGDMA content obtained using Dielectric Spectroscopy at 30 °C. The range of frequency values reported in Fig. 2a only captures the high-to-mid values below which polarization effects at electrode-electrolyte interface start to dominate. It is seen that at lower frequencies the conductivity is independent of frequency denoting the bulk or d.c. conductivity ($\sigma_o$). In the high frequency region, the conductivity values progressively rise beyond a critical frequency of $\omega_c$ due to correlated ionic motions and depend on the host media microstructure as well as temperature. Such behavior has also observed universally in several glassy and solid polymer electrolytes reported in the literature. The solid line fits in Fig. 2a represent the jump-relaxation model proposed by Jonscher[29] for thermally activated processes that was mathematically

reproduced by Cramer et al.[30] as a power-law expression: $\sigma = \sigma_o [1 + (\tau\omega)^p]$. Here $\tau$ is the timescale for dielectric relaxation that depends upon the coulombic interactions between the polymer chains and mobile ions. The power law exponent p denotes the ratio between the forward hop to backward hop relaxation time. The variation of $\tau$ and p with different PEGDMA content ($\phi$) is represented in Fig. 2b. It is seen that the ion hopping relaxation time ($\tau$) increases rapidly beyond $\phi = 40\%$. This can be attributed to a transition of conduction mechanism from bulk motion of oligomers to ion hopping processes along the EO links in the percolated network. The evidence of this is the transition from a mixed diglyme-network to a single-phase solid electrolyte observed from the glass transition behavior at and beyond $\phi = 40\%$. Furthermore, the p-value is seen to changes from >1 to less than unity at the same fraction. Previously in the literature, solid polymer electrolytes based on PEO have been reported to exhibit $p < 1$ due to the cationic nature of the ion transport where Li$^+$ ions hop through coordination with EO molecules, however, at elevated temperatures as well as for liquid electrolytes $p$ is seen be higher than 1. This argument similarly validates our proposed idea that all diglyme molecules are 'tightly' bound by the percolated network beyond $\phi = 40\%$.

Supplementary Fig. 6 reports the d.c. conductivity obtained from the plateau region of the frequency dependent conductivity as a function of inverse temperature. The continuous lines here represent the Vogel–Fulcher–Tammann (VFT) model given by, $\sigma_o = A \exp(-E_a/R(T-T_o))$, provides relationship between the ion transport rate and temperature. Here, A is a pre-factor related to the overall number of charge carrier, $E_a$ is the activation energy

and $T_o$ denote the characteristic temperature below which the ion transport ceases to take place. The excellent fits provided by the VFT model confirms the absence of any temperature induced chemical or morphological changes under the measurement conditions. The conductivity values obtained at 30 °C are reported as a function of ϕ in Fig. 2c. It is seen that from ϕ = 0 to 40% the conductivity decreases by around one order of magnitude, but thereafter, quickly drops off by several orders of magnitude. Operation of cells created using the membrane with ϕ > 40% would require increased operating temperatures, similar to the many families of solid polymer electrolytes previously reported in the literature[20,31,32]. Figure 2d reports the activation energy obtained from the VFT fits. It is evident that the activation energy for ion hopping at first increases gradually as the composition ϕ = 40%, then rises sharply thereafter. This implies that ion transport through the membrane occurs by dominantly liquid processes and consistent, with the Tg results discussed in the previous section, switch to bulk solid-like polymer behavior as the PEDGMA network becomes more effective in constraining local motion of the oligoether segments. Thus, it is apparent that the molecular interactions between the oligomers and polymer network segments (polyethers are known to interact strongly with PEG and even more strongly with the acrylate segments in the networks[27]), ultimately plays a crucial role in regulating largescale ion transport processes in the materials. Our hypothesis is that at some optimal composition near 40% PEGDMA, the constraints to motion provided by the network is just strong enough to inhibit large length scale transport that drives hydrodynamic instability, but weak enough to allow local liberations of the oligoether to enable ion transport.

In absence of forced convections, electrodeposition is a diffusion-limited process such that ion transport rate at every potential difference should be a function of ionic conductivity. However, at intermediate voltage differences, the ion transport rate should reach a maximum value, known as the limiting current density, which can be calculated as $J_{lim} = zFDc/\delta$, where $z$ is the charge, $F$ is Faraday constant, $c$ is bulk concentration, $D$ is ion diffusivity and $\delta$ is the diffusion layer thickness. At higher voltages, the anion migration rate exceeds the diffusion rate causing a breakdown of electroneutrality in a region near the electrode-electrolyte interface resulting in creation of a space charge region. It has been experimentally observed as well as predicted from Nernst-Planck Theory that a large electric field near the electrodes generates a convective flow field driving the ions across the space charge region, thereby producing ion fluxes in excess of the limiting current to yield what's termed 'over-limiting' conduction in liquid electrolytes[33–36]. Since, the ion migration rate under these conditions is set by the strength of the electroconvective flow, rather than by ion diffusion, the transition to overlimiting conductance results in several unwanted events at the electrode, including rampant dendritic growth on the electrodes and electrolyte degradation as a result of parasitic reactions with the resultant high-surface area electrodeposits. These instabilities are absent in all solid-state electrolytes because convection of any kind is suppressed.

**Hydrodynamic stability during electrodeposition**. Supplementary Fig. 7 reports current density as a function of voltage for our membranes of different PEGDMA content. The measurements were performed in symmetric (Li||Li) two electrode cells. The voltage was scanned in a staircase progression from 0 to 5 V vs. Li/Li+, and the resultant steady-state current was recorded. For electrolytes with lower PEGDMA contents (ϕ = 0–20%), the current density measured at constant voltage and normalized by $J_{lim}$ is observed to rapidly diverge, without any obvious plateau,

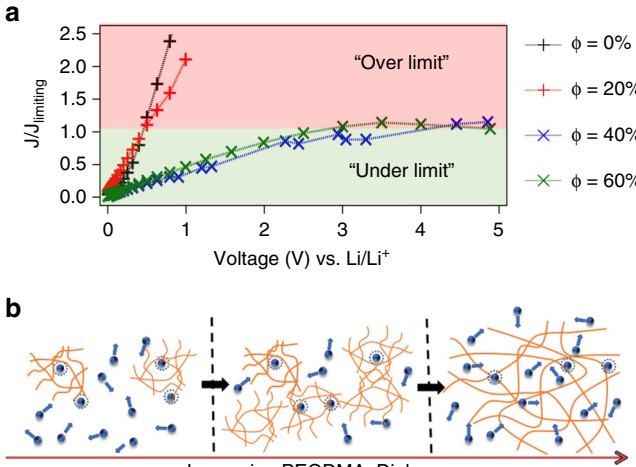

**Fig. 3** Hydrodynamic stability during electrodeposition. **a** Normalized I-V curves for membranes with different PEGDMA fractions (Φ). Here, the y-axis is normalized by the limiting current density. **b** Cartoon showing the evolution of the crosslinked polymer architecture with increasing PEGDMA content. As Φ increases, the fraction of "free" diglyme decreases in the networks and PEGDMA is dynamically coupled to the cross-linked polymer network in a singular percolated structure

implying that very strong convective processes are present in such cells, allowing over-limiting conduction to dominate. This observation can be contrasted with what is seen at higher PEGDMA contents (see Fig. 3a and Supplementary Fig. 6), where a clear plateau region is observed, and the current density appears pinned by the diffusion limiting current, even up to quite high voltages. Thus, for PEGDMA content ϕ ≥ 40%, the membranes are able to completely suppress electroconvective instability.

We believe that this feature of the materials arises from the fact that the PEGDMA polymer network and associated liquid electrolyte exist as a single percolating solid at higher (40–100%) PEGDMA content. A cartoon summarizing the observed effect is provided as Fig. 3b. The architecture of the cross-linked PEGDMA network in the solid polymer interphase membrane is shown. With diglyme in excess, the physically wet membrane contains a mixture of free- and network-associated diglyme chains. As the PEGDMA content is increased, the balance shifts to diglyme chains that are fully associated with network segments and as such are unable to move independent of the membrane. As a consequence, the liquid electrolyte behaves electrokinetically as part of the solid electrolyte membrane. It is therefore interesting that the oligomer-polymer interactions that were shown earlier to regulate microscale thermodynamics of the mixtures also control macroscale electrokinetics. Based on these observations, we select membranes with a PEGDMA content ϕ = 40% as ideal materials for designing Solid Polymer Interphase (SPI) that simultaneously exhibit liquid and solid-like characteristics.

**Morphological stability and electrochemical performances**. To characterize macroscopic morphological evolution at the anode during electrodeposition, an in-house built visualization cell was utilized. The cell contained dual-lithium metal rods as the electrodes coated with the SPI and liquid electrolyte of 1 M EC: DMC LiPF$_6$ in the interelectrode space, filling the center of the tube. Electrodeposition was visually recorded under an optical microscope using a current density of 4 mA cm$^{-2}$. Electrodeposit morphologies for both coated and uncoated electrodes were observed at regular intervals up to 1 h, as shown in Fig. 4a. For

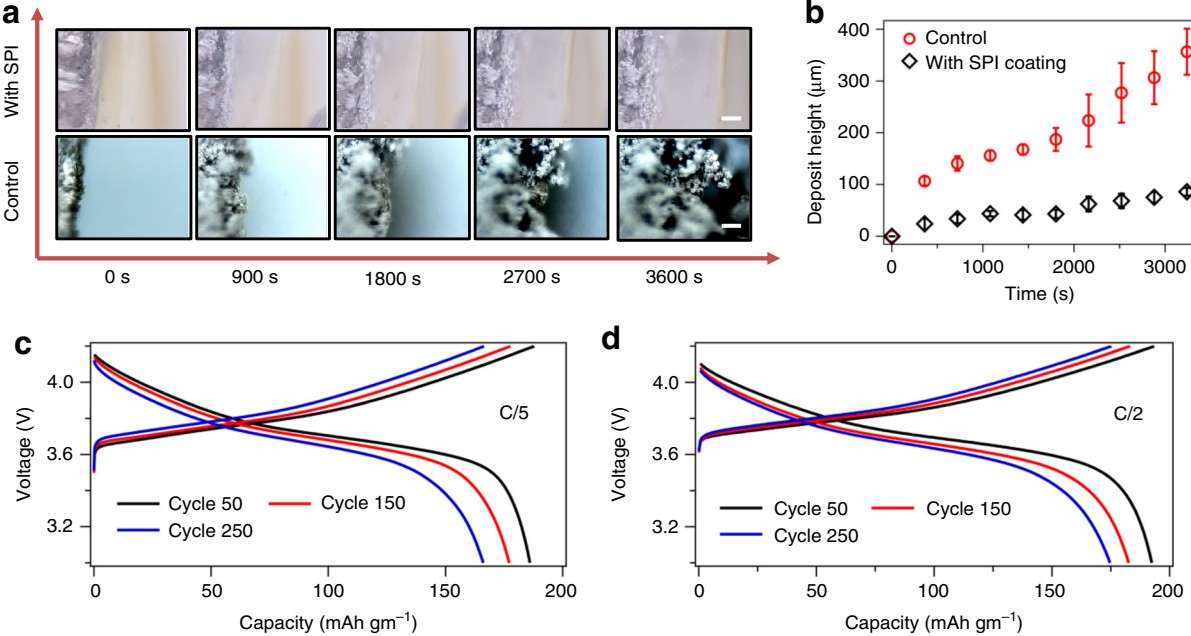

**Fig. 4** Morphological stability and electrochemical performance. **a** Snapshots of the Li/electrolyte interface from time-resolved optical visualization experiments. Snapshots in the first row are results for lithium coated with the synthesized solid polymer interphase; the second row shows results with the uncoated lithium electrode. In both cases the electrolyte is 1 M LiPF$_6$ in EC/DMC (50/50) and the current density is 4 mA cm$^{-2}$. The lithium metal anode is on the left side of each micrograph, while the counter Li electrode is on the right side (not shown here). In the first row, the polymer layer is not in focus. The scale-bars represent 100 μm. **b** Average height of the Li electrodeposit as a function of time. **c, d** Voltage profiles for Li||NCM batteries, the lithium anode is coated with the solid polymer interphase for C-rates of C/5 and C/2 respectively. The electrolyte used is 0.6 M LiTFSI, 0.4 M LiBOB 0.05 LiPF$_6$ in EC/DMC

the control electrolyte, the electrodeposit morphology is clearly "mossy" and rough as compared to the case with the solid polymer interphase. Quantitative information of electrodeposit morphology is reported in terms of the average height of the deposits versus time in Fig. 4b. In a linear regression of the data, the growth rate of the pristine lithium deposits with the control was found to be ~90 nm s$^{-1}$, while that with the SPI was found to be ~20 nm s$^{-1}$. It is also seen that the error bars on the control is significantly higher than that of the SPI, which further confirms the heterogeneities and roughness induced by the deposition without the SPI. Additionally, we evaluated the stability of electrodeposition for polymer coatings with varying contents of the PEGDMA contents using a galvanostatic strip-plate measurements in a symmetric lithium cell using a constant current density of 0.5 mA cm$^{-2}$. As seen in Supplementary Fig. 8, the batteries using $\phi = 20$ and 40% show stable cycling for at least 1000 cycles, while at higher PEGDMA contents, the overpotential is seen to rise rapidly or sever noise is observed in the voltage profiles. This indicates that the electrochemical properties are optimum at $\phi = 40\%$, and beyond that the poor conductivity and rigidity destabilizes the electrodeposition.

We evaluated battery impedance using various thicknesses of the SPI coating on the lithium metal anode in a symmetric cell configuration. It can be seen in Supplementary Fig. 9a, the bulk and interfacial impedances decrease with decreasing thicknesses of the SPI. We believe, the polymer coating is processable to very low thicknesses, useful for achieving very high volumetric energy density for practical applications. However, for the purpose of this report, we limit the thickness of the SPI coating to 100 μm range. We investigated the chemical stability of the lithium electrode incorporating the SPI coating using impedance spectroscopy measurement as a function of time shown in Supplementary Fig. 9b. It is observed that the interfacial impedance remains essentially unchanged which indicates that the lithium metal does not undergo side reactions in presence of the artificial interphase.

Additionally, we performed post-mortem analyses of the electrodeposited lithium with and without the solid polymer coating shown in Supplementary Fig. 10. In this experiment we deposited Li at the rate of 1 mA/cm$^2$ for one hour and observed the surface under scanning electron microscope. It was seen that without the polymer layer, the deposits are rough and mossy, while the polymer film enables compact electrodeposition morphology. Supplementary Fig. 11 reports the CE of cells with and without the SPI coating (with $\phi = 40\%$) as a function of cycle number. Results were obtained using an asymmetric coin cell, with pristine Li electrode and stainless-steel counter electrode with/without the SPI coating. In the control case, the cell cycles stably at above 80% CE for around 30 cycles, before the CE drops and the cell fails. In contrast, the coin cells fabricated with the SPI coating exhibited >80% CE for at least 100 cycles. It is clear that the SPI can promote long-term electrodeposition stability. The SPI coated on a lithium foil was paired with a Nickel Cobalt Manganese Oxide (NMC-622) intercalating cathode to investigate the electrochemical characteristics of the SPI material. An electrolyte composed of 0.6 M LiTFSI, 0.4 M LiBOB 0.05 LiPF$_6$ in EC/DMC (1/1 by wt.), which has been reported previously[37] to stabilize cell operation at high potentials was used for this part of the study. Figure 4c, d reports the voltage profiles at a rate of C/5 and C/2, respectively. The capacity and coulombic efficiencies as a function of cycle numbers are reported in Supplementary Fig. 12a, 12b. Here, where 1C corresponds to a current density of 2 mA cm$^{-2}$. It can be seen that at both C-rates the initial capacity is approximately 100 mAh gm$^{-1}$, which eventually increases to ~185 mAh gm$^{-1}$ and ~155 mAh gm$^{-1}$ for C/5 and C/2 respectively. Significantly, the CE is seen to remain close to 100% throughout the battery operation. This indicates that the observation is related to the activation of the electrode surface for reversible electrochemical reactions. It can be seen at both C-rates that the discharge capacity retention even after 250 cycles of operation is more than 90%. Also, in comparison to the full-cell cycling results for a control cell based on the same electrolyte

(Supplementary Fig. 12b), the cell comprising of the polymeric coating show improvement in capacity retention. We performed full-cell impedance analysis of the SPI case at different cycle numbers, as shown in Supplementary Fig. 13, where we found that at with the battery cycling there is a significant drop of resistance due to equilibration of ion transport channels through the polymer coating at the anode. This behavior also coincides with the rise of capacity during battery cycling due to the same reasons.

Motivated by these excellent full cell results, we wondered whether the SPI might also work as a standalone solid-state electrolyte for room temperature lithium metal batteries. We report 'strip and plate measurements' of a symmetric lithium cell in Supplementary Fig. 14, where the SPI ($\phi = 20$ and 40%) were sandwiched between Li metals. Interestingly, it can be observed in the solid-state polymer ($\phi = 40\%$) that resistance of the battery is low and there is no progressive rise in the overpotential or short-circuit for at least 2000 h even at room temperature operation. However, in case of the gel-like polymer ($\phi = 20\%$), the battery fails by a soft' short-circuit indicated by an abrupt fluctuation in the voltage profile. In comparison, the liquid electrolyte operation leads to short-circuiting by a sudden drop in the voltage. Clearly, it is seen that at the optimum PEGDMA content ($\phi = 40\%$), the battery overcomes the issues of solid-state batteries without much sacrifice in the overall conductance as achievable in liquid electrolytes. Furthermore, Supplementary Fig. 15 shows that even without any special efforts to optimize the composition of the SPI, a thicker version (~400 μm) of the material sustains stable cycling of an all solid-state Li||NCM battery based on a 50 μm Li foil anode and 2 mAh/cm$^2$ NCM cathode employed in the previous investigation (Fig. 4) in which the SPI is used in tandem with a liquid carbonate electrolyte. A key finding is that integration of LiBOB as a salt additive in the cross-linking step, yields an essentially all-ether, PEG-based polymer electrolyte that exhibits extended electrochemical stability at potentials above typical values where polyethers are conventionally regarded as prone to oxidative degradation. The source of this stability is tentatively attributed to anionic clusters formed by LiBOB at the electrolyte-cathode interface, which appears to facilitate desolvation of Li+ ions as they intercalate into the cathode. A more detailed understanding of these features of the electrolytes is the subject of ongoing studies.

## Discussion

In conclusion, we have designed cross-linked single-phase polymer networks based on polyether chemistry and illustrated their use as solid polymer interphases for room temperature lithium batteries. We show that interactions between a high boiling point solvent (bis(2-methoxyethyl) ether) with network segments are sufficient to couple the solvent to the network, causing the entire ensemble to exhibit micro- and macroscale transport characteristics similar to a solid-state electrolyte. In particular, it is reported that at low contents of a PEGDMA polymer network former, distinct glass transition temperatures (Tg's) are observed for the cross-linked PEGDMA and free oligoether components, while at higher PEGDMA contents, a single Tg is seen. Thus, at a critical content of 40% PEGDMA, the materials behave as single-phase soft solids in which the barrier to ionic transport in the oligoether are low enough to produce high ionic conductivity, but at the same time, the oligoether is prevented from exhibiting large-scale convective motions by interactions with the network chain segments. As an example of the latter characteristic, we show that coatings of the networks on a Li substrate produce liquid-like interfacial resistance yet are able to completely suppress the hydrodynamic instability known as electroconvection up to voltages as high as 5 V (~200 times the thermal energy, RT/F). We demonstrate the potential of these single-phase networks

as solid-electrolyte interphases on lithium metal anodes. We show that the ability of such networks to simultaneously ensure good interfacial ion transport, limit electrolyte access to the anode, and to flex and stretch to accommodate volume change during cycling of the anode facilitates stable lithium metal battery cycling by inhibiting dendritic growth and parasitic electrode-electrolyte side reactions. As a final demonstration of the utility of the materials, we report preliminary results, which show that the material can be used either as Li metal anode coatings—deployed in tandem with a liquid electrolyte, or as all solid-state polymer electrolytes to enable stable and high CE room temperature cycling of Li||NCM cells that utilize thin Li anodes and high-loading NMC cathode.

## Methods

**Fabrication of crosslinked polymer network and coated lithium**. PEGDMA (Mn = 750), Diglyme and Lithium Nitrate were purchased from Sigma Aldrich. All chemicals were thoroughly dried before usage. The PEGDMA and diglyme were mixed in different ratios as required, however the LiNO$_3$ content was maintained at Li:EO = 0.1 for all the samples. The mixture was thoroughly mixed to obtain a uniform solution. After addition of 4 wt.% of a photoinitiator methyl benzoylformate (MBF), the solution was casted on a desired substrate and exposed to UV light (VMR UVAC 115 V ~60 Hz 254/365 nm) for 20 min. After the reaction, the membranes were utilized as is for characterizations.

The solid polymer interphase was formed using the same procedure, however the reaction was carried out on a flat piece of lithium metal anode in an Argon-filled glove-box.

**Material characterization**. The molecular structuring in the glassy electrolytes were studied using attenuated total reflectance−Fourier transform infrared spectroscopy (ATR-FTIR) on a Nicolet iS10 FTIR spectrometer (Thermo Fisher Scientific) equipped with a deuterated triglycine sulfate (DTGS) detector and a SMART iTR diamond ATR accessory. Melting transitions were then investigated using DSC on a DSC Q2000 (TA Instruments) at a scan rate of 10 °C min$^{-1}$.

**Electrochemical characterization**. Ionic transport in the bulk and at the interface in this system was studied using conductivity and impedance measurements using a Novocontrol N40 broadband spectrometer fitted with a Quarto temperature control system. The samples were sandwiched between two gold-plated blocking electrodes. The I-V analysis were done using staircase voltammetry where each voltage steps comprise of 20 s using Maccor battery testers.

2030 coin-type cells were assembled in a glovebox (MBraun Labmaster) with NCM cathode (2 mA cm$^{-2}$) as the cathode and lithium foil (Alfa Aesar) as the anode. The solid-polymer coated lithium was paired with the NCM cathode, and the in a liquid electrolyte comprised of LiBOB (Oakwood Chemicals), LiTFSI (Sigma-Aldrich) and LiPF$_6$ (Sigma-Aldrich) salts in an Ethylene Carbonate/ Dimethyl Carbonate (Sigma Aldrich) mixture.

The CE tests were performed in a cell configuration of lithium anode with or without the solid polymer coating paired against a stainless-steel counter electrode. The electrolyte utilized was 1 M EC: DMC LiPF$_6$. In this measurement a fixed amount of lithium was plated onto the stainless-steel electrode and stripped back, such that the ratio of stripped and plated lithium determined the CE for each cycle.

The direct visualization experiment was done using two lithium rod-type electrodes in a tube-like visualization cell[38].

## Data availability

All datasets generated and analyzed during the current study are available from the corresponding author (LAA) on reasonable request.

## Code availability

All computer codes used for data analysis are available from the corresponding author (LAA) on reasonable request.

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

## Acknowledgements

This work was supported by the Department of Energy Basic Energy Sciences Program through Award DE-SC0016082. Electron microscopy and XPS analysis were performed in facilities supported by the Cornell Center for Materials Research with funding from the NSF MRSEC program (DMR-1719875).

## Author contributions

S.C. and L.A.A. conceived the idea and designed the experiments. S.C., S.S. and D.V. performed the soft matter characterizations and electrochemical measurements. S.C., D.V. and P.B. performed the operando visualization measurements. S.C. and A.W. performed the hydrodynamic analyses. Y.D. did the electron microscopy. All authors contributed in writing the paper.

## Competing interests

The authors declare no competing interests.
