## [Peer Review File · Nature Communications]

Reviewers' comments:

Reviewer #1 (Remarks to the Author):

This manuscript describes the design of polymer gel electrolytes to mitigate several challenges associated with the use of Li metal electrodes. The article is well-written and for the most part, the authors' conclusions are fully supported by the data presented. The authors use in situ light microscopy to demonstrate that the polymer gel membranes reduce dendritic growth. Electrochemical characterization of lithium battery half cells (NMC vs Li) employing these polymer membranes show reversible, stable cycling. The characterization of the polymer chemistry and the material properties was thorough. However, the vinyl conversion is not reported. This is a key property that defines network structure, material properties, and potentially chemical/electrochemical stability. Can the authors estimate vinyl conversion from their FTIR data?

However, authors' conclusions could be further strengthened by addressing the following issues and questions regarding electrochemical characterization:

1. Include scale bars in Figure 4a. It is difficult to discriminate the various regions in the micrographs, e.g. where is the current collector, dense Li metal deposit, Li dendrites, and polymer layers? Labels or annotations of these micrographs would be helpful.
2. In Figure 4b, dendrites that are 80um high grow in the SPI-coated substrate within one hour. While this is an improvement over the control, how significant is this improvement? Can the authors put this in context relative to other solid-state and interfacial design strategies (with references)? Given that any lithium battery is going to require electrolyte regions that are on the order of 20 um or less to limit ASR and minimize cell volume, it seems that limiting the dendrites to 80 um is not a significant improvement in stabilizing the electrode morphology.
3. It would also be enlightening to see similar dendrite growth characterization in membranes of varying compositions. Does dendrite growth slow as the membrane becomes stiffer (less glyme/more PEGDMA)?
4. In Figure S8 a & b, the capacity of the cell increases over the approximately first 20 cycles. The author suggest on page 10 of the manuscript that the increase in capacity is due to the liquid electrolyte wetting the SPI. However, 20 cycles at C/5 and C/2 require approximately 200 hours and 80 hours, respectively. Wetting is a surface phenomenon and should establish equilibrium in time frames that are orders of magnitude shorter. It is unclear what the authors are suggesting by this statement and further clarification is required.
5. The electrochemical characterization of the NMC-Li cells would benefit from a control without the SPI layer in order to understand the effect of the additional resistance of the membrane on the rate performance of the cell. To understand the role of the SPI in enabling stable cycling and high capacity retention, cycling performance of the control without the SPI at identical conditions should also be included.
6. In Figure S9 the coulombic efficiency of is consistently above 100% over the entire 25 cycles. Why?

The manuscript would also benefit from more exhaustive referencing. Supporting references should be cited for the following discussions:

- C=O FTIR analysis (second paragraph, page 4)
- Gordon-Taylor relationship (last sentence, page 5)
- Discussion of over-limiting conduction (first paragraph, page 8)
- Also, the first full sentence on page 8 (starts with "At intermediate voltages...") is incomplete.

Reviewer #2 (Remarks to the Author):

The data obtained from the present work are acceptable while the paper is not well documented and there are some typos. Following comments need to be considered.

1. In Introduction, the authors reviewed about alkaline metal anodes, but in this manuscript, only Li metal anode was used. Therefore, the introduction should focus/include on Li metal anode.
2. More detail characterization of the polymer used is needed. For the degree of polymerization, there is no evidence showing cross-linked polymer network.
3. In cell test, LiPF₆ salt was used in liquid part while LiNO₃ in polymer part so that characterization of polymer was performed only in LiNO₃ salt. When both salts are used, electrochemical performance can be differently. The effects are not discussed
4. For Supplementary Fig. 2, the change of FTIR spectra ~ wavenumber 950 /cm should be explained? Two small peaks disappear with the increase of PEGDMA, why?

Reviewer #3 (Remarks to the Author):

This manuscript reports a PEGDMA-based polymer electrolyte mainly used as interphase coatings on a metal anode in a liquid electrolyte (one brief example as solid-state electrolytes in Fig. S9), which is considered to address all failure modes of a Li metal anode. This is an interesting topic. However, this reviewer does not recommend its publication of this manuscript.

1. Many soft polymeric coatings have been used to stabilize the lithium metal-electrolyte interface and enable high-rate and high-capacity lithium metal cycling (e.g., J. Am. Chem. Soc. 2018, 140, 11735-11744; Angew. Chem. 2018, 130, 1-6; Adv. Energy Mater. 2018, 8, 1701482). Compared with these reported works, the reviewer finds lack of enough novelty in this study.
2. To better understand the role of the SEI layer in stabilizing the electrolyte-Li interface, the ionic and electronic conductivities and the mechanical properties of the PEGDMA-based SEI layer should be measured and discussed.
3. For illustration, the authors only report a single thickness of approximately 100 μm of the interphase coatings. However, this thickness is too thick for the high-energy density batteries. How about thin coatings?
4. As for T_g value of $\phi = 20\%$ that does not follow the Gordon-Taylor relation, in page 5, it is explained as to "form a dense and swollen percolated network of polymer chains". However, there is no evidence in FTIR pattern or any visible proof. Please clarify the inner logic.

5. The authors claim that their polymer electrolytes can be used as solid-state electrolytes for solid-state batteries, but the supporting data are too limited. Figure S9 in the supporting information shows cycling data for just 25 cycles, which is not enough. Other properties such as mechanical properties of the polymer membrane and the cycling performance of Li|polymer|Li symmetric cells should be provided.

6. The coulombic efficiency (Fig. S9) in the solid-state cell Li|NCM looks like higher than 100%. What's the reason behind?

Response to Reviewer Comments

Reviewer #1 (Remarks to the Author):

This manuscript describes the design of polymer gel electrolytes to mitigate several challenges associated with the use of Li metal electrodes. The article is well-written and for the most part, the authors' conclusions are fully supported by the data presented. The authors use in situ light microscopy to demonstrate that the polymer gel membranes reduce dendritic growth. Electrochemical characterization of lithium battery half cells (NMC vs Li) employing these polymer membranes show reversible, stable cycling. The characterization of the polymer chemistry and the material properties was thorough. However, the vinyl conversion is not reported. This is a key property that defines network structure, material properties, and potentially chemical/electrochemical stability. Can the authors estimate vinyl conversion from their FTIR data?

Response: In the revised manuscript (pg. 4, final paragraph), we complement our analysis of the strong FTIR band at 1700 cm^{-1} , which is attributed to the C=O bond associated with methacrylate groups in the coatings, with a more detailed analysis of the much weaker band at 1650 cm^{-1} associated with vinyl groups. This analysis shows that the band is substantially weaker in the cross-linked membranes than in the un-crosslinked PEGDMA and completely disappears at low ϕ . The findings are consistent with our analysis of the C=O band that the membranes are formed by the well-known radical-initiated cross-linking reaction of PEGDMA and show that the cross-link density increases at higher ϕ .

Reviewer: However, authors' conclusions could be further strengthened by addressing the following issues and questions regarding electrochemical characterization:

1. Include scale bars in Figure 4a. It is difficult to discriminate the various regions in the micrographs, e.g. where is the current collector, dense Li metal deposit, Li dendrites, and polymer layers? Labels or annotations of these micrographs would be helpful.

Response: We thank the reviewer for the careful scrutiny of the manuscript. We have added scale-bar in the figure and detailed descriptions of micrographs in the figure legends.

Reviewer: 2. In Figure 4b, dendrites that are 80um high grow in the SPI-coated substrate within one hour. While this is an improvement over the control, how significant is this improvement? Can the authors put this in context relative to other solid-state and interfacial design strategies (with references)? Given that any lithium battery is going to require electrolyte regions that are on the order of 20 um or less to limit ASR and minimize cell volume, it seems that limiting the dendrites to 80 um is not a significant improvement in stabilizing the electrode morphology.

Response: We now see that the y-axis label in Figure 4 is a potential source of confusion in understanding the significance of the results presented in figure 4. What is plotted in the figure is not the height of the dendrites, but rather the thickness of the Li electrodeposit. This change has been made in the revised manuscript. It means that rather than inferring effectiveness of the membranes from the absolute height of the deposit, which in fact would be expected to increase approximately linearly with time if the Li deposits in a dendrite-free morphology, it is the variation of the thickness (error bars) with location that reports on the roughness of the electrodeposit morphology. The results in the figure shows that the membranes not only substantially increase the density of the Li electrodeposits relative to the control case, but that they also reduce the deposit roughness from very large values (order $100\text{ }\mu\text{m}$) at high Li throughputs to lower levels. We stress here that this is achieved without the aid of normal force at the interphase that would be induced by pressure ($> 500\text{ psi}$) imposed during normal cell assembly. To complete this discussion, we've also included in the revised manuscript results from SEM analysis of unmodified and SPI-coated Li electrodes in Li symmetric cells cycled at 1 mA/cm^2 and an electrodeposit capacity of 4 mAh/cm^2 . Since

our objective here is to understand the added effect of the coating, the same commercial Celgard separator and electrolyte were used for the control and membrane-coated Li electrodes. The results reproduced in

Figure 1 below reveal a similar flattening effect of the electrodeposit morphology when the SPI is employed.

Fig. 1: SEM micrographs showing the electrodeposition on metallic anode for one hour at the rate of $1\text{mA}/\text{cm}^2$. The left image is for a bare electrode while the right is for the electrode with a layer of the solid polymer coating. In both cases the electrolyte utilized was 1M LiPF_6 in EC/DMC. Both scale bars are $100\mu\text{m}$.

Reviewer: 3. It would also be enlightening to see similar dendrite growth characterization in membranes of varying compositions. Does dendrite growth slow as the membrane becomes stiffer (less glyme/more PEGDMA)?

Response: The requested analysis is presented in the revised manuscript. Prompted by the reviewer's question, we performed additional galvanostatic strip-plate experiments to examine the stability of Li coated with SPI with varying compositions. All measurements were performed at a moderate current density of $0.5\text{mA}/\text{cm}^2$ as recommended by the reviewer. The results reported as Supplementary Fig. S7 in the revised manuscript show that membranes with the lower and intermediate ϕ values (i.e. $\phi = 20\%$ & 40%) are the most effective in enhancing the cycling stability of the Li anode. For membranes with higher ϕ ($\phi = 60\%$ & 80%) the overpotential is high at the selected current density, which leads to erratic cycling of the cells. The result is consistent with the arguments presented in the paper for selecting the membranes with $\phi = 40\%$ as optimized systems for more in-dept studies. We fully expect, however, that changes in the PEGDMA molecular weight employed in making the SPI and the electrolyte solvent hosted in the pores of the material will affect the optimal composition.

Reviewer: 4. In Figure S8 a & b, the capacity of the cell increases over the approximately first 20 cycles. The author suggest on page 10 of the manuscript that the increase in capacity is due to the liquid electrolyte wetting the SPI. However, 20 cycles at $C/5$ and $C/2$ require approximately 200 hours and 80 hours, respectively. Wetting is a surface phenomenon and should establish equilibrium in time frames that are orders of magnitude shorter. It is unclear what the authors are suggesting by this statement and further clarification is required.

Response: Upon closer scrutiny of the cycling data we see that the initial capacity rise is not accompanied by changes in the Coulombic efficiency, which remains steady and high. This indicates that the observation is related to the activation of the electrode surface for reversible electrochemical reactions. The slow increase in the capacity has been previously observed in ether based gels and solid electrolytes and associated with the formation of percolated ionic pathways over cycles for full utilization of the active materials.

Reviewer: 5. The electrochemical characterization of the NMC-Li cells would benefit from a control without the SPI layer in order to understand the effect of the additional resistance of the membrane on the rate performance of the cell. To understand the role of the SPI in enabling stable cycling and high capacity retention, cycling performance of the control without the SPI at identical conditions should also be included.

Response: The reviewer raises a valid point. In the revised manuscript, we have compared the full cell cycling of a Li||NMC cell at C/2 rate, with and without the polymer coating.

Reviewer: 6. In Figure S9 the coulombic efficiency of is consistently above 100% over the entire 25 cycles. Why?

Response: We apologize for this error. In the earlier manuscript, the inverse of the CE was reported in Figure S9. This error has been corrected in the revised manuscript.

Reviewer: The manuscript would also benefit from more exhaustive referencing. Supporting references should be cited for the following discussions:

- C=O FTIR analysis (second paragraph, page 4)
- Gordon-Taylor relationship (last sentence, page 5)
- Discussion of over-limiting conduction (first paragraph, page 8)
- Also, the first full sentence on page 8 (starts with “At intermediate voltages...”) is incomplete.

Response: We have added the relevant citations as recommended by the reviewer and also corrected the mentioned statement.

Reviewer #2 (Remarks to the Author):

The data obtained from the present work are acceptable while the paper is not well documented and there are some typos. Following comments need to be considered.

Reviewer: 1. In Introduction, the authors reviewed about alkaline metal anodes, but in this manuscript, only Li metal anode was used. Therefore, the introduction should focus/include on Li metal anode.

Response: We welcome this comment., however we believe the polymer coatings/electrolytes concept is straight-forward enough to be applied to other reactive metals, including sodium which have similar modes of failure mechanisms as Li. Although for brevity, we only focus on Li in this first study, we thought that a broader introduction would introduce a wider cross-section of readers to the results reported.

Reviewer: 2. More detail characterization of the polymer used is needed. For the degree of polymerization, there is no evidence showing cross-linked polymer network.

Response: The un-crosslinked PEGDMA is a simple liquid with no elasticity and no ability to sustain a tensile strain without flow. In the revised manuscript we performed small-amplitude oscillatory elongation experiments in a tensile testing device to quantify the development of elasticity in the materials as a function of PEGDMA content. The results reported in Supplementary Figure S3 not only show that the materials exhibit high tensile elastic moduli (as high as 10MPa, which is approximately 20-times higher than that of a physically cross-linked PEG polymer), but that the elastic moduli are essentially independent of oscillation frequency. This latter feature is a well-known, tell-tale sign of a covalently cross-linked network as it signifies a paucity of energy dissipation due to relaxation of the deformed polymer chain segments. These results provide even more conclusive support than the IR data in Supplementary Figure S2, where it is seen that the peak at 1650cm^{-1} for the C=C bond in the uncrosslinked PEGDMA significantly decreases in the high PEGDMA containing networks while they disappear when present at lower content. The information is added and discussed on page 5, first paragraph, of the revised manuscript.

Reviewer: 3. In cell test, LiPF₆ salt was used in liquid part while LiNO₃ in polymer part so that characterization of polymer was performed only in LiNO₃ salt. When both salts are used, electrochemical performance can be differently. The effects are not discussed

Response: The liquid electrolyte was solely used to wet the cathode surface for better contact with the solid polymer electrolyte. Our broader interest is to explore the concept of a two-phase electrolyte that can be sustained in a battery cell. In such a system a solid polymer electrolyte suitable to solve issues related to the anode is coupled with a liquid electrolyte suitable for high voltage stability. Importantly, LiNO₃ is sparingly soluble in carbonate electrolytes, thus it can be easily limited in the polymer interphase. LiPF₆ and LiBOB are used to enhance oxidative stability of the SPE in full-cell battery studies employing nickel-rich NMC cathode chemistries.

Reviewer: 4. For Supplementary Fig. 2, the change of FTIR spectra ~ wavenumber 950 /cm should be explained? Two small peaks disappear with the increase of PEGDMA, why?

Response: The reviewer makes a good point in identifying a detail of the “fingerprint region” of the FTIR spectra that we missed in the earlier manuscript. Although it is difficult to identify these peaks with confidence, what is clear is that a transition from multiple discrete and weaker IR bands to a single stronger IR band occurs for $\Phi \geq 20\%$. This finding is consistent with the transition to a solid-like state beyond $\Phi = 40\%$ as it is expected as molecular segments belonging to the polymer and diglyme chains trapped in the networks become more confined and therefore less free to rotate and precess. These points are now discussed in greater detail on page 4 of the revised manuscript.

Reviewer #3 (Remarks to the Author):

This manuscript reports a PEGDMA-based polymer electrolyte mainly used as interphase coatings on a metal anode in a liquid electrolyte (one brief example as solid-state electrolytes in Fig. S9), which is considered to address all failure modes of a Li metal anode. This is an interesting topic. However, this reviewer does not recommend its publication of this manuscript.

Reviewer: 1. Many soft polymeric coatings have been used to stabilize the lithium metal-electrolyte interface and enable high-rate and high-capacity lithium metal cycling (e.g., J. Am. Chem. Soc. 2018, 140, 11735-11744; Angew. Chem. 2018, 130, 1-6; Adv. Energy Mater. 2018, 8, 1701482). Compared with these reported works, the reviewer finds lack of enough novelty in this study.

Response: The reviewer's point is well taken. However, we note that all of the reports mentioned by the reviewer were published after submission of the present work. We note further that despite the large volume of studies that have begun to appear in this area, the present report stands apart in at least four important aspects:

- (i) Unlike previous reports involving polymer coating on lithium metal anodes, our current system can be utilized as a solid-state electrolyte.
- (ii) The current work takes a fundamental approach in decoupling the effects of the morphological as well as hydrodynamic instabilities using different experimental techniques.
- (iii) We demonstrate that using classical ion transport and polymer physics approach, one can understand the effect of the solid-polymer architecture on its electrochemical properties.
- (iv) Our work also demonstrates a facile and novel strategy of enabling ether-based polymers as anodic interfaces in high voltage batteries.

Reviewer: 2. To better understand the role of the SEI layer in stabilizing the electrolyte-Li interface, the ionic and electronic conductivities and the mechanical properties of the PEGDMA-based SEI layer should be measured and discussed.

Response: The ionic conductivities and the mechanical properties have been provided in Supplementary Figures S6 and S3, respectively. In addition, we performed impedance analysis of the SPI coatings on Li in the presence of a liquid carbonate electrolyte and report these results as Supplementary Figure S9. It can be seen from that figure that consistent with other evidence showing that the SPI forms a stable interphase that does not undergo further polymerization/parasitic side reactions in contact with a Li anode, the impedance remains remarkably stable with time.

Reviewer: 3. For illustration, the authors only report a single thickness of approximately 100 μm of the interphase coatings. However, this thickness is too thick for the high-energy density batteries. How about thin coatings?

Response: Our goal in this first submission was not to build a fully functioning battery, but simply to illustrate a simple and potentially general approach for creating elastic interphases on Li anodes. The reviewer makes a valid point nonetheless that it would be good to see how the SPI perform at other thicknesses. In the revised manuscript, we have carried out additional studies to evaluate performance of SPIs with thicknesses as low as 25 μm .

Reviewer: 4. As for Tg value of $\phi = 20\%$ that does not follow the Gordon-Taylor relation, in page 5, it is explained as to "form a dense and swollen percolated network of polymer chains". However, there is no evidence in FTIR pattern or any visible proof. Please clarify the inner logic.

Response: Support for our hypothesis that the SPI are composed of individual PEG chains jointed together to form a covalently cross-linked, macroscopic material actually comes from multiple complementary sources. In the revised manuscript, the formation of the percolated network is clarified using oscillatory elongational rheology, ionic conductivity, and transport activation energy measurements. The results for the elongational rheology experiments are reported in Supplementary Figure S3. We note that whereas the

un-crosslinked PEGDMA is a simple liquid with no long-range connectivity, and therefore no elasticity and ability to sustain a tensile strain without flow, irrespective of the PEGDMA content, the SPI exhibit high levels of tensile strength. The tensile elastic moduli are observed to rise with PEGDMA content to as high as 10MPa, which is approximately 20-times higher than that of a physically cross-linked PEG polymer, a clear sign that the constituent molecules are covalently linked on macroscopic length scales. Additionally, the elastic moduli are seen to be essentially independent of oscillation frequency. This feature is a well-known, tell-tale sign of a covalently cross-linked network as it signifies a paucity of energy dissipation due to relaxation of the deformed polymer chain segments and supports our hypothesis that the individual chains are connected to form a well-developed percolated network of PEG ionic conductors. These results are also consistent with those deduced from the conductivity and ion transport measurements in Fig 2. It is seen that there is a sudden drop in conductivity as well as increased activation energy beyond $\phi=20\%$. While these features do not p observed due to formation of the percolating PEGDMA network. The electrochemical and electro-kinetic observation is also seen to follow ta similar pattern. Furthermore, we discuss in the revised manuscript regarding the presence of multiple peaks in the FTIR spectra in the wavenumber region below 1000cm^{-1} upto $\phi=20\%$, indicative of multiple modes of relaxations that is absent beyond that, due to the formation of a dense network.

Reviewer: 5. The authors claim that their polymer electrolytes can be used as solid-state electrolytes for solid-state batteries, but the supporting data are too limited. Figure S9 in the supporting information shows cycling data for just 25 cycles, which is not enough. Other properties such as mechanical properties of the polymer membrane and the cycling performance of Li|polymer|Li symmetric cells should be provided.

Response: As recommended by the reviewer, we have added the cycling of Li|polymer|Li cell in Supplementary Figure S13 (also given below). It is seen that the overpotentials are low and do not rise rapidly for at least 1000 hours of cycling at $0.5\text{mA}/\text{cm}^2$. Other properties of the polymer membrane like the mechanical modulus and ionic conductivity at different temperature is provided in Supplementary Figure S3 and S6, respectively.

Fig 2: Li|polymer|Li cycling at $0.5\text{mA}/\text{cm}^2$ with each half cycle being one hour.

6. The coulombic efficiency (Fig. S9) in the solid-state cell Li|NCM looks like higher than 100%. What's the reason behind?

Response: We apologize for this error. In the earlier manuscript, the inverse of the CE was reported in Figure S9. This error has been corrected in the revised manuscript.

Reviewers' comments:

Reviewer #1 (Remarks to the Author):

The authors have substantially improved the manuscript through their revisions. However, I do not feel that some of my original critiques have been adequately addressed.

2. I now understand that Figure 4B is the electrodeposit thickness. However, the y-label is "Dendrite Height," and the caption states "average height of dendrite as a function of time." This is highly confusing since the average electrodeposit thickness is actually being plotted. Moreover, based on the rebuttal, the error bars represent variation in thickness among several locations, from which you can infer a surface roughness or nonuniformity. The significance of the error bars should be included in the manuscript.

Since electrodeposit thickness is being plotted, Figure 4b suggests that the electrodeposited Li is not dense even with the polymer electrolyte. 4 mAh/cm² capacity should have thickness of approximately 20 micrometers, but Figure 4b is showing approximately 80 micrometers. How is this reconciled – especially considering Figure S10, which shows a relatively smooth surface morphology.

4. I still don't understand what is happening with the capacity rise in Figure S12. The explanation in the main body of the manuscript is, "This indicates that the observation is related to the activation of the electrode surface for reversible electrochemical reactions." It is not clear what this means and why it doesn't happen in the control cell with standard liquid electrolyte. In the rebuttal letter, it is mentioned that this is associated with formation of percolated ionic pathways that are previously observed for these gels. References are needed in the manuscript. Also, that brings up the question what is the liquid electrolyte in the cathode porosity? If liquid electrolyte is absorbed in the gel electrolyte, what liquid is filling the cathode porosity? Was excess liquid electrolyte used to infiltrate the cathode?

Reviewer #2 (Remarks to the Author):

The manuscript is well revised and addressed all my concerns. Therefore I recommend for consideration of publication.

Reviewer #4 (Remarks to the Author):

This manuscript reports an oligomers containing polymer coating on Li anode to address instability problems of Li anode. I reviewed the comments from the reviewers and the responses from the authors, I find that there're still some issues need to be clarified before publication. The advantages and novelty of this work need to be clarified as polymers coating have been commonly used to modify Li anode (related published papers need to be cited and discussed). Although the polymer coating in this work can be utilized as a solid-state electrolyte, the solid battery can only work 25 cycles. What's the improvements of this work compared with liquid battery and solid battery?

The structure of the polymer coating is unclear. Please provide more visible proofs.

More proofs or exhaustive references should be given to illustrate the "molecular interactions between the oligomers and polymer network segments" and "interactions between bis(2-methoxyethyl) ether and network segments".

More exhaustive references should be included in this manuscript, such as, the explanation in page 4,

about "the C=O bond shifts to lower wave number with increasing PEGDMA content", "the change in "figure-print region" of FTIR" and so on.

There are some typos. Supplementary Fig. 8 miswrite as Supplementary Fig. 9. Moreover, the current density of symmetric lithium cell tests is 0.5 mA/cm² in the manuscript. However, the current density of the same experiments is described as 1 mA/cm² in the corresponding annotations of the Supplementary Fig. 8 (the first Supplementary Fig. 9). Please confirm which one is correct.

In the symmetric lithium cell tests (the first Supplementary Fig. 9), the overpotential and cycling performance of the batteries with 20% PEGDMA are all better than that of 40% PEGDMA. However, the authors declared that the optimum one is the polymer with 40% PEGDMA. It is hard to understand.

Why the T_g value of the polymer with 20% PEGDMA does not follow the Gordon-Taylor relation (in page 5)? Please provide visible proof.

Reviewer #1 (Remarks to the Author):

The authors have substantially improved the manuscript through their revisions. However, I do not feel that some of my original critiques have been adequately addressed.

2. I now understand that Figure 4B is the electrodeposit thickness. However, the y-label is “Dendrite Height,” and the caption states “average height of dendrite as a function of time.” This is highly confusing since the average electrodeposit thickness is actually being plotted. Moreover, based on the rebuttal, the error bars represent variation in thickness among several locations, from which you can infer a surface roughness or nonuniformity. The significance of the error bars should be included in the manuscript.

Response: We thank the reviewer for pointing this out. The axis has been changed to deposit height since the entire electro-deposit is being plotted as a function of time. The error bars are an indication of non-uniformity/surface roughness as noted by the reviewer. The control case is visibly rougher and non-uniform compared to that with SPI as can be inferred from the much smaller error bars in the latter case. We have added further discussions in the revised manuscript.

Since electro deposit thickness is being plotted, Figure 4b suggests that the electrodeposited Li is not dense even with the polymer electrolyte. 4 mAh/cm² capacity should have thickness of approximately 20 micrometers, but Figure 4b is showing approximately 80 micrometers. How is this reconciled – especially considering Figure S10, which shows a relatively smooth surface morphology.

Response: The reviewer is correct. Under the conditions of the experiment, the thickness of an electrodeposited Li layer that achieved the theoretical density of the bulk metal would be 20 μm, as opposed to the 80μm measured with the coating and >350 μm measured without the coating. The difference between the theoretical (20 μm) and measured (80μm) electrodeposit thickness arises from the absence of a separator in the visualization cell. Under these conditions, Li electrodeposits grow in the absence of external pressure and as such are not as compact as can be achieved in coin-cell measurements in which pressure is applied via a separator. It is a limitation of the in-house built optical visualization cell that will be remedied in a next generation setup designed to control and quantify the effect of external pressure on the electrodeposit growth rate.

4. I still don't understand what is happening with the capacity rise in Figure S12. The explanation in the main body of the manuscript is, “This indicates that the observation is related to the activation of the electrode surface for reversible electrochemical reactions.” It is not clear what this means and why it doesn't happen in the control cell with standard liquid electrolyte. In the rebuttal letter, it is mentioned that this is associated with formation of percolated ionic pathways that are previously observed for these gels. References are needed in the manuscript. Also, that brings up the question what is the liquid electrolyte in the cathode porosity? If liquid electrolyte is absorbed in the gel electrolyte, what liquid is filling the cathode porosity? Was excess liquid electrolyte used to infiltrate the cathode?

Response: The capacity rise was hypothesized to originate from improved interfacial contact between the electrolyte and the cathode during cycling. The better contact was thought to produce a lower interfacial resistance over time facilitating more efficient utilization of the active material in the electrode. To validate this hypothesis, we performed additional impedance measurements for the full cells at different stages of cycling (*i.e.* before cycling, at cycle #25, and cycle #50). The results are reported in Fig. S13. We fitted the results with the circuit model shown in the inset of Fig. S13 and observed that while all of the cell resistances in the model varied during cycling, the resistance of the cathode electrolyte interphase changes the most – it decreases by more than a factor of three in the first 50 cycles and by a factor of approx. two between cycles 25 and 50.

Reviewer #2 (Remarks to the Author):

The manuscript is well revised and addressed all my concerns. Therefore I recommend for consideration of publication.

Response: We thank the reviewer for favorable recommendation regarding publication.

Reviewer #4 (Remarks to the Author):

This manuscript reports an oligomers containing polymer coating on Li anode to address instability problems of Li anode. I reviewed the comments from the reviewers and the responses from the authors, I find that there're still some issues need to be clarified before publication. The advantages and novelty of this work need to be clarified as polymers coating have been commonly used to modify Li anode (related published papers need to be cited and discussed). Although the polymer coating in this work can be utilized as a solid-state electrolyte, the solid battery can only work 25 cycles. What's the improvements of this work compared with liquid battery and solid battery?

Response: The reviewer raises a valid point. To answer the reviewer's question, we performed additional battery cycling studies using the polymer coating as a solid electrolyte (without addition of free liquid or separator) and performed galvanostatic strip-plate cycling measurements in a symmetric Li||Li cell at a moderate current density of 0.5 mA/cm^2 . The results are reported in Figure S14 and reproduced below. It can be clearly seen that the cells with PEGDMA content = 40% exhibit over 2000 hours (i.e. > 1000 cycles) of strip and plate cycling, without any sign of failure. In comparison cells using either the gel-like electrolyte (PEGDMA content = 20%) and liquid electrolyte fail by dendrite-induced shorts in < 250 hours. Thus, it is clear that the proposed methodology for designing solid-state cross-linked polymer electrolytes can result in significant improvement in long term cycling stability (similar to expectation in a solid electrolyte), without significant compromise in the interfacial conductance or ambient operation (similar to liquid electrolytes).

Fig. S14: Comparison of symmetric cell cycling using different electrolyte mechanics: (a) Symmetric lithium metal battery cycling at a rate of 0.5 mA/cm^2 with each half cycle being one hour. The electrolyte utilized was solid state polymer comprising of polymer network and diglyme, with PEGDMA content of 40% and with the salt LiNO_3 (Li:EO = 0.10). The thickness of the solid polymer electrolyte was $\sim 400 \mu\text{m}$. The inset shows the expanded voltage profiles at different time of cycling **(b)** cycling using same conditions as a, however the electrolyte used with PEGDMA content of 20% that has gel-like texture; **(c)** results for liquid electrolyte of 1 M LiPF_6 in EC/DMC (without any separator), instead an PTFE O-ring was used.

The structure of the polymer coating is unclear. Please provide more visible proofs.

Response: The chemical structure of the polymer coating has been provided in Figure 1 of the revised manuscript.

More proofs or exhaustive references should be given to illustrate the “molecular interactions between the oligomers and polymer network segments” and “interactions between bis(2-methoxyethyl) ether and network segments”.

Response: We have provided the relevant reference in the revised manuscript.

More exhaustive references should be included in this manuscript, such as, the explanation in page 4, about “the C=O bond shifts to lower wave number with increasing PEGDMA content”, “the change in “figure-print region” of FTIR” and so on.

Response: In the revised manuscript, we have provided references on the same.

There are some typos. Supplementary Fig. 8 miswrite as Supplementary Fig. 9. Moreover, the current density of symmetric lithium cell tests is 0.5 mA/cm² in the manuscript. However, the current density of the same experiments is described as 1 mA/cm² in the corresponding annotations of the Supplementary Fig. 8 (the first Supplementary Fig. 9). Please confirm which one is correct.

Response: We thank the reviewer for pointing out the errors and typos. The current density is 0.5 mA/cm², we have corrected that in the revised manuscript.

In the symmetric lithium cell tests (the first Supplementary Fig. 9), the overpotential and cycling performance of the batteries with 20% PEGDMA are all better than that of 40% PEGDMA. However, the authors declared that the optimum one is the polymer with 40% PEGDMA. It is hard to understand.

Response: There is an interplay between decreasing conductivity and increasing storage modulus as we go from 0% to 100% crosslinker in the coating. The optimum was chosen to be 40% because it is the point where the system transitions to a one phase material as evident from the DSC experiments as well as the composition with optimum conductivity and mechanical properties. To validate this, we assembled all solid-state symmetric cells with 20% PEGDMA and 40% PEGDMA membranes (Figure S14). The results confirm that the material containing 40% PEGDMA is exceptional both as an artificial SEI and as a solid-state electrolyte.

Why the T_g value of the polymer with 20% PEGDMA does not follow the Gordon-Taylor relation (in page 5)? Please provide visible proof.

Response: 20% PEGDMA shows two T_g's that are between the corresponding T_g's of the pure states. Since, the PEGDMA is crosslinked, there is inadequate mixing with diglyme liquid, thus we observe two T_g's representing blending heterogeneities. Similar behavior has been observed in the case of partially miscible polymer blends that have interfacial mixing. We have provided addition discussion and relevant references.

REVIEWERS' COMMENTS:

Reviewer #1 (Remarks to the Author):

The authors have thoroughly addressed my critiques. I believe the manuscript is much more complete and should be considered for publication.

Reviewer #4 (Remarks to the Author):

The manuscript is well revised and addressed all my concerns. Therefore I recommend for consideration of publication.